# SARS-CoV-2 and the Risk Assessment Document in Italian Work; Specific or Generic Risk Even If Aggravated?

**DOI:** 10.3390/ijerph18073729

**Published:** 2021-04-02

**Authors:** Luigi Cirrincione, Venerando Rapisarda, Walter Mazzucco, Rosanna Provenzano, Emanuele Cannizzaro

**Affiliations:** 1Department of Health Promotion, Maternal and Child Health, Internal Medicine and Medical Specialties “Giuseppe D’Alessandro”, University of Palermo, 90127 Palermo, Italy; luigicirrincione@gmail.com (L.C.); walter.mazzucco@unipa.it (W.M.); 2Department of Clinical and Experimental Medicine, Occupational Medicine, University of Catania, 95123 Catania, Italy; vrapisarda@unict.it; 3Department of Juvenile and Community Justice, 93100 Caltanissetta, Italy; rosanna.provenzano@giustizia.it

**Keywords:** COVID-19, risk assessment document, SARS-CoV-2, occupational medicine, decree 81/2008, work, law, prevention and protection measures

## Abstract

In December 2019, several cases of atypical pneumonia were detected in Wuhan city, Hubei province, inland China. The initial outbreak was of considerable size first in China subsequently spread to the rest of the world. Immediately after the epidemic (which according to the World Health Organization had risen to pandemic status), the problem of whether or not to update the occupational risk assessment arose, also considering how the biological risk from SARS CoV-2 should be understood: specific or generic. To this end, we conducted a literature review to identify national health legislation and policies, examining how Italy has addressed the COVID-19 emergency in occupational health planning, in order to develop considerations on the need to update the Risk Assessment Document following the pandemic status. The data that emerged from the review of current legislation allowed us to conclude that the risk from SARS-CoV-2 is in most work activities to be understood as a generic or aggravated generic risk, requiring the employer to apply and control the preventive measures suggested by health authorities to contain the spread of the virus.

## 1. Introduction

In December 2019, several cases of atypical pneumonia were detected in Wuhan city, Hubei province, inland China. The first patients, it would later be discovered, had shown symptoms in early December, or even in mid-November. However, it was on the last day of the year 2019, in a 41-year-old patient admitted 5 days earlier, that doctors officially identified a new virus called SARS-CoV-2 as the cause of this atypical pneumonia [1].

The initial outbreak was of considerable size first in China and then spread to the rest of the world; on 11 March 2020, the World Health Organization (WHO) declared that the new SARS-CoV-2 coronavirus infection could be considered a pandemic [2].

Epidemiological data on the progress of the pandemic, and Coronavirus disease or more commonly COVID-19 (from Corona VIrus Disease-19), are constantly evolving, updated daily by the World Health Organization [3].

SARS-CoV-2 has been shown to be less lethal overall than previously isolated viruses of the same family, with a mortality rate of 5.6% in Europe and 3.1% in the United States (data as of August 2020) [4,5,6].

In Italy, from the beginning of the epidemic until 1 July 2020, the date of the last update of the ISS (Italian Higher Institute of Health) databases to prepare the report “Impact of the COVID-19 epidemic on total mortality in the resident population in the period January–May 2020,” approximately 240,000 positive COVID-19 cases were reported. The number of COVID-19 cases reported in Italy peaked in March and then gradually declined during the summer months, resuming with the second pandemic wave in October [7].

Current evidence suggests that SARS-CoV-2 is spread both directly, through direct and close contact with infected persons, through secretions from the mouth and nose (saliva, respiratory secretions, or droplets), and indirectly through objects or surfaces contaminated with the same secretions [8].

Italy was the first country in the Euro–Mediterranean area to dramatically experience the SARS-CoV-2 pandemic. On February 21st the first “secondary” outbreak was detected, in which transmission did not affect only people coming from areas at risk. The first identified Italian patient with COVID-19 was identified in Codogno (Lodi), not being linked to known outbreaks. [1]

In relation to its progress, entire cities and countries have implemented lockdowns to reduce further spread of the virus. A key feature of this pandemic response is that individual countries chose their own strategies without coordinating with their neighbors [9].

The Italian state, like other countries in the world, progressively implemented increasingly restrictive measures to prevent the spread of the contagion, following the guidelines of the World Health Organization [10].

These restrictive measures included social distancing, community and fellowship strategies [11], research into possible non-pharmaceutical interventions to mitigate the infection [12] up to and including a national blockade from 8 March 2020, i.e., a total freeze of all non-essential manufacturing and commercial activities until June 2020 [13].

For production activities and essential services, which could not closed, the “Shared protocol for the regulation of measures to combat and contain the spread of the virus in the workplace” was published, signed on 14 March 2020 [14], which provided, among other things, the use, where possible, of the working method defined as “agile or smart-working” to minimize contact between employees of the same company [15].

Subsequently, when all production and commercial activities reopened, containment measures divided into organizational, environmental and personal were applied.

Organizational measures are general measures for the containment and management of the COVID-19 epidemiological emergency imposed by the competent authorities with the aim of minimizing the probability of being exposed to this new virus.

Among the organizational measures proposed to address the COVID-19 pandemic, it was considered of paramount importance to:Minimize the entry of visitors into the workplace by limiting or restricting access to all company personnel, including employees.Prevent people with obvious flu-like symptoms from entering company premises by having their body temperature taken, including all staff, customers, suppliers and external collaborators, not allowing anyone with a temperature above 37.5 °C to enter company premises.Inhibit access to traveling personnel from SARS-CoV-2 high-risk areas, initially defined as “red zones,” or high-class provinces based on the distribution of standardized cumulative incidence rates.Reduce the number of workers within individual confined spaces by taking advantage of all possible home-based agile work arrangements and workplace shifts.Create, where possible, two or more closed and independent workgroups, to be alternated every 14 days to work alternately in the company and in smart-working [16].

Among environmental measures aimed at reducing the risk of transmission of SARS-CoV-2 infection through contact with infected individuals, or through contact with inanimate or contaminated objects, equipment, and surfaces, the greatest emphasis has been placed on cleaning and sanitizing work environments, using for this purpose both solutions containing chemical agents such as ethanol (62–71%) or hydrogen peroxide (0.5%), for adequate contact time, and providing increased ventilation of enclosed spaces following their use [17]; physical means such as ultraviolet irradiation or the use of ozone that have demonstrated the ability to maintain high levels of disinfection for extended periods [18,19,20].

In November, following the increase in contagion due to what has been unanimously recognized as the “second pandemic wave”, new restrictive measures and selective limitations were issued through the Decree of the President of the Council of Ministers (now called DPCM) of 3 November 2020 in the Italian regions according to the different epidemic trends in the territory [21].

## 2. Purpose

Soon after the outbreak (which had reached the level of a pandemic according to the World Health Organization), the question arose as to whether or not the risk assessment needed to be updated.

In the field of health and safety at work, the assessment of health and safety risks is the fundamental tool that allows the employer to identify the prevention and protection measures and to plan their implementation, improvement and control in order to verify their effectiveness and efficiency. In this context it is possible to confirm the safety measures already in place or make changes to improve them in relation to technical and/or organizational innovations introduced in the field of safety.

The assessment of risks to health and safety is of fundamental importance among the general measures of protection, being the prerequisite for the whole system of prevention.

The main purpose of risk assessment is to determine whether or not the preventive measures taken are adequate so that risks can be controlled before harm occurs. In order to achieve a comprehensive risk assessment, it is necessary to use a participatory approach; that is, to involve staff in identifying and understanding problems in the work environment and then be able to implement improvements in the safety and health of the worker and the work facility involved.

The evaluation of the risks must therefore cover all the risks, according to the modifications introduced by the European Community, and must consequently result in a document (called in Italy the document of evaluation of risks or DVR) containing at least:A report on the assessment of occupational health and safety risks, specifying the criteria adopted for the assessment.The identification of protective and preventive measures and individual protective equipment.The program of measures considered appropriate to ensure that safety levels are improved over time.

The risk assessment can be used to outline the actions necessary to eliminate and/or minimize the potential for harm (active and passive prevention and worker protection).

In the specific case of Italian legislation, following the lines of work suggested by Legislative Decree 81/2008, the following actions are possible (whose possible effectiveness can be compared through the hierarchy of risk controls):Hazard elimination;Changing the circumstances and causes of hazardous situations that cannot be eliminated in order to control them and be able to prevent the potential risk;The elimination of harm and/or its reduction to low levels of severity.

The purpose of our work was therefore to analyze the current legislation on occupational safety, the ministerial decrees issued to prevent the spread of the pandemic, to understand how and whether risk management should be re-evaluated.

## 3. Materials and Methods

We conducted a literature review to identify national health legislation and policies to examine how Italy has addressed the COVID-19 emergency in workplace health planning. Information from the review and current legislation was used to develop considerations about the need to update the risk assessment document following pandemic status.

Evidence gathered from the review was found primarily in the form of legislation, including the Commission (EU) Directive, Council Directive, WHO Status Report, Prime Minister’s Decree (PMD), legislative decrees, and national health protocols.

In particular, more attention has been paid to Directive 2000/54/EC of the European Parliament and of the Council of 18 September 2000, Commission Directive (EU) 2019/1833 of 24 October 2019, Commission Directive (EU) 2020/739 of 3 June 2020, Council Directive 89/391/EEC of 12 June 1989, Situation Report 12 of February 2020 issued by the WHO, to the various Decrees of the President of the Council of Ministers that have followed throughout the pandemic phase until the end of 14 January 2021 containing the measures to combat and contain the emergency from COVID-19, to the Consolidated Safety Act 81 of 9 April 2008 and the shared protocol that regulates the measures to combat and contain the spread of COVID-19 virus in the workplace.

### Most Relevant Regulations

Regulatory Requirements:Legislative Decree no. 81 of 9 April 2008—Consolidated law on safety in the workplace;Decree Law no. 6 of 23 February 2020—Urgent measures for containment and management;The COVID-19 epidemiological emergency, which was subsequently supplemented by the Legislative Decree of 25 March 2020, containing additional urgent measures to address the COVID-19 epidemiological emergency;Ministry of Health Circular 22 February 2020—Ministry of Health Circular. COVID-19, new guidance and clarification;International Public Health Emergency Clarification of the World Health Organization of 30 January 2020 by which the provisions of the International Health Regulations were activated and the subsequent declaration of 11 March 2020 by which the outbreak of COVID-19 was assessed as a “pandemic” in view of the levels of diffusivity and severity reached globally;“Shared Protocol for Regulating Measures to Combat and Contain the Spread of COVID-19 Virus in the Workplace” dated 14 March 2020, which was subsequently supplemented with the Shared Protocol for Combating the Virus in the Workplace dated 24 April 2020;DPCM 26 April 2020, containing guidance for the start of “Phase 2”;Italian National Institute for Insurance against Accidents at Work (now called INAIL) technical paper of April 2020: on the possible remodeling of measures to contain SARS-CoV-2 infection in the workplace and prevention strategies;Ministry of Health Circular 29 April 2020 Operational guidance on the activities of the competent physician as part of measures to combat and contain the spread of the SARS-CoV-2 virus in the workplace and community;Memorandum of Understanding “Operational Guidelines to Ensure the Smooth Running of State Final Examinations 2019/2020”, signed between the Ministry and the unions on 15 May 2020;Decree Law No. 33 of 16 May 2020 Additional urgent measures to address the epidemiological emergency from COVID-19;DPCM 17 May 2020, Implementing provisions of Decree-Law No. 19 of 25 March 2020, on urgent measures to cope with the epidemiological emergency from COVID-19, and Decree-Law No. 33 of 16 May 2020, on additional urgent measures to cope with the epidemiological emergency from COVID-19;Act 22 May 2020, No. 35. Conversion into law, with amendments, of Decree-Law No. 19 of 25 March 2020, containing urgent measures to address the epidemiological emergency from COVID-19;D. Act No. 83 of 30 July 2020, Urgent measures related to the expiration of the COVID-19 epidemiological emergency declaration approved on 31 January 2020;DPCM of 7 August 2020 “Further provisions implementing Decree-Law No. 19 of 25 March 2020, on urgent measures to cope with the epidemiological emergency from COVID-19, and Decree-Law No. 33 of 16 May 2020, on further urgent measures to cope with the epidemiological emergency from COVID-19”;Note of the Ministry of Labor and Social Policies—Ministry of Health No. prot. 13 of 4 September 2020 and 28877 of 4 September 2020, Circular of the Ministry of Health of 29 April 2020 containing “Operational indications relating to the activities of the competent doctor in the context of measures to combat and contain the spread of the SARS-CoV-2 virus in the workplace and the community”. Updates and clarifications, with particular regard to frail workers and female workers;Legislative Decree No. 125 of 7 October 2020 “Urgent measures related to the extension of the declaration of a state of epidemiological emergency by COVID-19 and for the operational continuity of the COVID alert system, as well as for the implementation of Directive (EU) 2020/739 of 3 June 2020”;Ministry of Health Circular Prot. No. 32850, 12 October 2020—COVID-19: guidance for duration and termination of isolation and quarantine;Legislative Decree of 19 October 2020 by the Minister for Public Administration, on “Measures for agile work in public administration in the period of the emergency”, published in the Official Gazette of the Italian Republic—General Series No. 268 of 28 October 2020;DPCM of 14 January 2021, Further provisions implementing Decree-Law No. 19 of 25 March 2020, converted, with amendments, by Law No. 35 of 22 May 2020, bearing “Urgent measures to cope with the epidemiological emergency from COVID-19”, Decree-Law No. 33 of 16 May 2020, converted, with amendments, by Law No. 74 of 14 July 2020, bearing “Further urgent measures to cope with the epidemiological emergency from COVID-19”, and Decree-Law No. 2 of 14 January 2021, bearing “Further urgent measures to cope with the epidemiological emergency from COVID-19”. Law No. 74 of 14 July 2020, entitled “Further Urgent Measures to Cope with the COVID-19 Epidemiological Emergency,” and of Decree-Law No. 2 of 14 January 2021, entitled “Further Urgent Provisions on the Containment and Prevention of the COVID-19 Epidemiological Emergency and the Calling of Elections for the Year 2021.”Legislative Decree 81/2008 in art. 268 of Title X (Exposure to biological agents) classifies biological agents into the following four groups (borrowed from European Directive 2000/54/EC):-Group 1 biological agent: an agent that is unlikely to cause disease in human subjects;-Group 2 biological agent: an agent that can cause disease in human subjects and pose a risk to workers; unlikely to spread in the community; effective prophylactic or therapeutic measures are usually available;-Group 3 biological agent: an agent that can cause serious illness in human subjects and poses a serious risk to workers; the biological agent can spread in the community, but effective prophylactic or therapeutic measures are usually available;-Group 4 biological agent: a biological agent that can cause severe disease in human subjects and poses a serious risk to workers and may present a high risk of propagation in the community; no effective prophylactic or therapeutic measures are normally available [22].
In October 2019, Directive (EU) 2019/1833 amended Annex III of Directive 2000/54/EC by adding a number of biological agents to various groups, including Severe Acute Respiratory Syndrome Coronavirus (SARS-CoV) and Middle Eastern Respiratory Syndrome Coronavirus (MERS-CoV); in relation to the risk reduction measures to be put in place, it is clear that it was essential to include SARS-CoV-2 in one of these groups [23,24].The new Commission Directive (EU) 2020/739 of June 3, 2020, in light of the latest available scientific evidence and clinical data, as well as the opinions provided by experts representing all Member States, in order to continue to ensure adequate protection of the health and safety of workers in the workplace, includes SARS-CoV-2 among the group 3 biological agents [25,26].

## 4. Discussion

As was easily foreseeable, the coronavirus emergency has generated a series of delicate interpretative questions which, in the field of labor law, also concern the application of the preventive discipline for the protection of health and safety at work provided for by Legislative Decree 81/2008.

On the contrary, if we look closely, the management of the current emergency, precisely because it is intimately linked to the protection of people’s health, risks creating real short circuits with the discipline of health and safety at work, subjected, as never before, to a tension that risks undermining some essential principles. Specifically, considerable uncertainties weigh on the prevention measures to be taken in production activities.

Immediately after the outbreak of the epidemic (which has risen to the level of pandemic according to the World Health Organization), the problem arose of whether or not to update the risk assessment, which, as foreseen by art. 29, paragraph 3, of Legislative Decree no. 81/2008, “must be immediately revised, according to the procedures” of art. 28, paragraphs 1 and 2, “on the occasion of changes in the production process or the organization of work that are significant for the health and safety of workers, or in relation to the degree of evolution of the technique, prevention or protection or following significant accidents or when the results of health surveillance show the need for it”, [27] with consequent updating of prevention measures. A risk assessment that Legislative Decree 81/2008 defines, in line with the principle contained in Framework Directive no. 89/391/EEC, as the “global and documented assessment of all risks to the health and safety of workers in the organization in which they work, aimed at identifying appropriate prevention and protection measures and drawing up a program of measures to ensure the improvement of health and safety levels over time” (art. 2, letter q; art. 28) [28].

In light of these predictions, the question then arose as to whether the risk of coronavirus infection should be taken into account by the employer by updating the risk assessment already carried out and the relevant document.

To answer the question of how to update the Risk Assessment Document, and what measures to adopt, it is necessary to consider how the current pandemic linked to the SARS-CoV-2 emergency poses a series of unprecedented problems in the workplace, to which it is necessary to respond quickly and effectively, in order to ensure that companies have adequate prevention measures against coronavirus infection to protect their workers (in cases where it is still possible to carry out the work activity, under the conditions most recently indicated by DPCM, also taking into account what can be found in the “Protocol shared between Government and social partners” of 14 March 2020).

Without wishing to go into the merits of the complex subject matter, given the proliferation of interpretations and the diffusion of tools of every form and content (guidelines, operational indications, protocols, etc.), it is necessary to clarify some legal elements on the provisions contained in the various DPCMs:-These are not measures aimed exclusively at businesses, but at all citizens; therefore, they are aimed at the general public and not just businesses;-As they are mandatory, they must also be applied in the field of health and safety at work, insofar as they provide for procedures for the protection of public health that affect the prevention and protection measures in force, supplementing what is imposed by the legislation in force;-Being an integral part of a document of undeniable legal value, they provide for mandatory measures for individuals, with reference to the penalties provided for by the criminal code in case of non-compliance, but do not modify or repeal acts having the force of law (such as, for example, Legislative Decree 81/2008).

The Consolidated Safety Act provides for the assessment of all risks present within the organization in which workers operate, i.e., the specific risks listed in the decree itself, which may be structural, instrumental, procedural and that the employer has designed and implemented in pursuit of its production purposes.

It is clear that the biological risk from SARS-CoV-2, which manifests itself through contact between people, can creep into the productive organizations in which people work. However, there is no doubt that, with the exception of some specific work activities, (such as those carried out in health services and hospitals, where the increase in infection rates in health workers could cause the collapse of the health system and a further worsening of the pandemic) [29,30]; in other cases, far from becoming a specific occupational risk, it is identified as a generic risk. This risk does not derive from the organization set up by the employer and does not necessarily manifest itself in it, but uses the organization and the complex system of personal relationships to manifest itself and spread, even if it comes from outside the organization itself: this is the case of a worker who becomes infected in an environment outside the company and, by going to work there, introduces the virus.

An interpretation that can help us finally is that which distinguishes the biological risk “intra-firm” from that of “extra-firm” origin.

With respect to the former, endogenous, the work environment entails a significant and qualified increase in the levels of exposure to a certain hazard and in the probability (or severity of the damage in the event) of the materialization of a certain risk, which will therefore be a risk specific to that work activity. On the contrary, in the face of a biological factor external to the productive organization in which the worker is inserted, the work activity is only an unqualified occasion for the occurrence of the event of damage to which that risk refers. The latter, in other words, is generic to the company’s activity, which is linked to it in terms of neutrality; the worker is therefore exposed to this risk, to a no lesser extent, even in the performance of other human activities outside the company’s premises.

In reality, on closer inspection, it would be more correct to base this distinction not so much on the internal or external origin of the risk factor, but rather on the fact that the work activity, intended both as working conditions and as a mere environmental context in which the service is performed, involves a level of exposure to risk higher than that “socially accepted” in the community to which the worker belongs. We are well aware of the limits to the application of a general clause as broad as the one just hypothesized, and in this regard we are helped by the definition of aggravated generic risk, but this clarification is necessary to include within the present reconstruction also cases that would otherwise have to constitute an exception.

If we broaden the discourse, it is clear that we cannot reach the paradoxical conclusion that every external factor affecting the company becomes an occupational hazard.

The issue of so-called exogenous risks is a serious one. For example, when a company sends one of its workers to a country with a high risk of malaria transmission, an aspect that is normally well known and foreseeable even in the light of the continuous updates provided by the Ministry of Foreign Affairs, one can well say that in the organization of that company, understood in a non-reified sense as a production project and as a set of rules that govern it, including the sending of workers abroad, this risk is intrinsic. Therefore, it is a risk of contagion when workers are sent to countries where other epidemics are known and foreseeable. Obviously, the employer will need to assess these risks and, if they cannot avoid them, will need to put in place all measures to counteract and reduce them.

In the case of the coronavirus, the situation is quite different; first, it is not a risk to one or more organizations, but to the entire social structure globally, which is why it has been defined as a pandemic by the WHO. Moreover, while in the case of malaria the risk could be avoided by not sending the worker abroad, or perhaps by having him or her collaborate with interlocutors in that country in “smart-working” mode, in the case of coronavirus this is not possible since the risk is always present; paradoxically, the worker may be considered safer within a company where effective precautionary measures have been taken than elsewhere.

On the other hand, if we reach the paradox whereby every employer classifies coronavirus risk as an occupational risk, we risk losing sight of what is actually an occupational risk.

All this must also be considered in those professional contexts that subject workers to particular conditions, such as those that expose them to particularly intense psychophysical stress, involving shift work, or night work, that recently considered by the International Agency for Research on Cancer (IARC) as probably carcinogenic to humans (Group 2A) [31], or that contemplate other contexts to be considered particular such as those in which social distancing is not feasible, see health technicians, professional team sports, etc. [32,33,34,35,36].

In these cases, far from considering SARS-CoV-2 as a specific occupational risk, it can be framed as an aggravated generic risk, which does not derive from the organization established by the employer or which must necessarily manifest itself in that organization, but rather uses the organization and the complex system of personal relationships to manifest itself and spread, coming from outside the organization itself [37].

In this case, in fact, the worker is more exposed, both in terms of intensity and frequency to the pandemic risk. Finally, although the notion of aggravated risk is not expressly contained in the law, it is unanimously recognized by jurisprudence with reference to a risk that, although common to all citizens, nevertheless constitutes an etiological link with work activity [38].

In light of what has been said so far, qualifying the risk arising from SARS-CoV-2 as an aggravated generic risk does not require the employer to develop a new scientific and technical approach to risk assessment and to the identification of precautions to be taken. The employer will proceed to the adaptation of those measures already identified (e.g., regarding the possible presence and influx of people, the maximum level of cleanliness and hygiene, the adoption and supply of the identified Personal safety protection devices or PPE) that reflect the protection indications already described in general conditions by the emergency regulations, the safety protocols and the national DPCM, to be adapted only to the specific work activity carried out.

Although the employer does not have to apply Title X of Legislative Decree 81/2008 on exposure to biological agents (since coronavirus is not a biological agent ontologically inherent to this organization, it is a biological and non-specific risk), he must therefore ensure the application of all prevention measures dictated by the Public Authority, evaluating how to adopt them in his company.

## 5. Conclusions

Therefore, it can be concluded that it will be those companies that deliberately use biological agents (e.g., microbiological research laboratories) or where there is a possibility of exposure inherent to the type of activity performed (health care facilities) that will need to review the DVR. All other activities where exposure to SARS-CoV-2 is not inherent to the type of activity performed, but derives solely from conditions specific to the epidemiological context, should be exempt. In the first production contexts, if the employer, with the collaboration of the prevention and protection service and the competent doctor, found a preventive inadequacy of his own safety system with respect to the new Coronavirus, he would certainly be obliged to update the DVR and to adopt those prevention and protection measures necessary to guarantee the control of exposure to this risk. On the contrary, in companies where the latter represents only a generic or aggravated risk, it will be sufficient for the employer to apply and monitor compliance with the preventive measures suggested by the health authorities to contain the spread of the virus, keeping track of it through an appropriate attachment.

This interpretation is also supported by the text of article 29, paragraph 3, of Legislative Decree 81/2008, which governs the hypotheses that give rise to the obligation to update the DVR. There are four hypotheses, namely:Changes in the production process or work organization that impact worker health and safety;Technological developments that allow for better prevention;The occurrence of significant incidents;Health surveillance results showing the need to update the document.

Therefore, among the causes from which the obligation to revise the DVR derives, environmental circumstances extraneous to specific company risks, such as the hypothesis of an epidemic, are not indicated. It therefore seems possible to exclude, also in the light of the cited provision, an obligation to update the document for those companies whose production process does not envisage activities that involve deliberate (or aggravated, even if accidental) exposure to biological risk, presenting only a generic risk of exposure.

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
