# Peer review of "SARS-CoV-2 and the Risk Assessment Document in Italian Work; Specific or Generic Risk Even If Aggravated?"

_ijerph, 2021, doi:10.3390/ijerph18073729_

Round 1

Reviewer 1 Report

Strengths

The document is very interesting, it is very important understand the risk from SARS-CoV-2 and the necessity of update the Risk Assessment Document to the safety of the employer

GENERAL ASPECTS:

The paper has a good structure, maybe it could be useful evaluating in the discussion what other countries do differently from Italy and if it was more effective than the Italian strategy.

I could only suggest reading two papers that might be useful to improve the introduction and discussion.

For the introduction I suggest reading “Time Effectiveness of Ultraviolet C Light (UVC) Emitted by Light Emitting Diodes (LEDs) in Reducing Stethoscope Contamination” Messina G., Fattorini M., Nante N., Rosadini D., Serafini A., Tani M., Cevenini G. International J. Of Environmental Research and Public Health, 13, 10, 940, 2016 to deepen the effectiveness of the ultraviolet irradiation.

In the discussion I might suggest the paper “Night work and quality of life. A study on the health of nurses.” Turchi V, Verzuri A, Nante N, Napolitani M, Bugnoli G, Severi FM, Quercioli C, Messina G Annali Ist Superiore Sanità, vol.55, No.2, 2019 to explore contexts that expose  employees to particularly intense psychophysical stress as the shift work.

Author Response

Thanks for your suggestions, we have followed your advice and inserted the bibliography indicated.

Reviewer 2 Report

The paper has specifically described the situation in Italy. Well done!

As the SARS-CoV2 virus is present generally all over the world the suggestion should be done that the best workplace preventive mesures should be supplied all over the world as well.

P.S. The sentences in the lines 339-342 are doubled!

Author Response

(The authors gave the same response as above.)

Reviewer 3 Report

The article is interesting and well written, however, there is a need to increase considerations on the general aggravated risk in healthcare professionals.
The bigliographic entry should be mentioned: Barranco R., Ventura F.: Covid-19 and health-care workers infection: an emerging problem. Medico-legal Journal 2020, 88(2): 65-66, 2020.

Author Response

(The authors gave the same response as above.)

Reviewer 4 Report

Overall, I do not see the main message that you are trying to deliver to us. The context leading up to what you really want to tell us is not clear, either. If the main point is the importance of the use of biological agents, we need more supporting sentences in terms of scientific evidence from past researches. What is the data you collected? How did you collect them? Any statistical analysis that supports your hypothesis? Any literature review (as in meta-analysis)?

The subsection 2.1 (most relevant regulations) needs to be put in the supplementary section as this does not seem to alter the context despite its detail. 

Author Response

Dear Reviewer Thank you for your advice.
I want to clarify that our work is a revision, for this reason it has no data and therefore it is not possible to carry out any statistical analysis.
The purpose of our study and the purpose of our work was therefore to analyze the regulations in force on safety in the workplace, the ministerial decrees issued to prevent the spread of the pandemic, to understand how and if the management of risks in the work-place should be reviewed.

Round 2

Reviewer 4 Report

Okay. I understand that this is a review article on the management of risks in the work-place in Italy. I would recommend that this article be shortened and sent as a letter/short communication rather than a full article if there is no statistical/systematic review involved in the contexts. Otherwise, there is no point to be corrected. 

Author Response

Very kind reviewer.
Thanks again for your kindness and your suggestions.
I would like to clarify that our work isn't absolutely a scientific article with data, or a systematic review; both of which require statistical analysis.
Ours work is a "literature review", specifically of the laws in force on safety at work, and protocols issued to prevent the spread of the pandemic; and like a literature review is structured in three parts: the introduction, the main body, and the conclusion.

For this reason, and in consideration of the work carried out, I do not consider can be shortened and sent as a letter or short communication.

best regard emanuele